# BotanicX-AI: Identification of Tomato Leaf Diseases Using an Explanation-Driven Deep-Learning Model

**DOI:** 10.3390/jimaging9020053

**Published:** 2023-02-20

**Authors:** Mohan Bhandari, Tej Bahadur Shahi, Arjun Neupane, Kerry Brian Walsh

**Affiliations:** 1Department of Science and Technology, Samriddhi College, Bhaktapur 44800, Nepal; 2School of Engineering and Technology, Central Queensland University, Norman Gardens, Rockhampton 4701, Australia; 3Central Department of Computer Science and IT, Tribhuvan University, Kathmandu 44600, Nepal; 4Institute for Future Farming Systems, Central Queensland University, Rockhampton 4701, Australia

**Keywords:** EfficientNetB5, eXplainable AI, GradCAM, tomato leaf diseases, LIME, deep learning

## Abstract

Early and accurate tomato disease detection using easily available leaf photos is essential for farmers and stakeholders as it help reduce yield loss due to possible disease epidemics. This paper aims to visually identify nine different infectious diseases (bacterial spot, early blight, Septoria leaf spot, late blight, leaf mold, two-spotted spider mite, mosaic virus, target spot, and yellow leaf curl virus) in tomato leaves in addition to healthy leaves. We implemented EfficientNetB5 with a tomato leaf disease (TLD) dataset without any segmentation, and the model achieved an average training accuracy of 99.84% ± 0.10%, average validation accuracy of 98.28% ± 0.20%, and average test accuracy of 99.07% ± 0.38% over 10 cross folds.The use of gradient-weighted class activation mapping (GradCAM) and local interpretable model-agnostic explanations are proposed to provide model interpretability, which is essential to predictive performance, helpful in building trust, and required for integration into agricultural practice.

## 1. Introduction

Early detection and control of crop disease is essential for farmers, stakeholders, and precision agriculture researchers to reduce the production losses. Current farm practices rely on visual identification of plant diseases by farm staff with the backup of specialists using additional resources and tools, such as microscopes [1]. However, agricultural professionals cannot constantly be present in the field to perform thorough monitoring, and farmers lack the expertise required to conduct the detection procedure [2].

Multispectral, RGB, and hyperspectral sensors have been used for crop-disease detection [3]. Recently, crop-disease detection utilizing a variety of image sensors has shown encouraging results when combining data-driven approaches, such as machine learning (ML) and deep learning (DL) [4]. Tomatoes are a commercially significant vegetable crop on a global scale, and various pathogens (viral, bacterial, and fungal illnesses [5,6,7,8]) that affect tomatoes have been identified [9].

A number of researchers have focused on the use of classification models in disease diagnosis. The majority of the suggested classifiers are developed and validated, with an emphasis on extracting deep features from images in order to categorize the foliage disorders [10,11]. For instance, in an experiment, Trivedi et al. [12] classified nine different kinds of tomato leaf diseases using a convolution neural network and a dataset of 3000 tomato leaves. They attained an accuracy of 98.49% using pre-processed and segmented tomato leaf images.

Although the existing deep-learning models for tomato-leaf-disease recognition achieved high accuracy on selected leaf image datasets, their interpretability and explainability are not sufficiently investigated to engender trust in using such models in practice. The eXplainable Artificial Intelligence (XAI) and DL algorithms that produce human-readable explanations for AI judgments lay the groundwork for imaging-based artificial-intelligence applications [13] in various domains, such as health informatics [14], computer vision [15], and many more.

Given that DL-based learning may autonomously extract features from an image without the need for human feature engineering, it is vital to explain the model’s output in cases when the XAI can enhance it. A few studies have anticipated XAI with DL models for the prediction of different subtypes of tomato leaf diseases (TLD) to include explanatory results [16]. Considering the high accuracy of DL models on TLD disease detection and the lack of explainability of the model’s output by existing works, this study aims to build a DL-XAI framework for TLD disease detection.

In this work, a TLD detection approach using a pre-trained DL model and an explainable solution highlighting the most significant portions of the foliage that contribute to disease categorization is proposed. The goal of the DL-XAI guided solution is to aid agricultural-based decision making. Additionally, the major contributions in this study are summarized as follows:
1.An explainable driven deep-learning framework for TLD recognition is developed using transfer learning with EfficientNetB5. The framework is named “(BotanicX-AI)”.2.GradCAM and LIME are utilized to provide explanation of the outcomes provided by the BotanicX-AI framework.3.The proposed study compares current pre-trained DL models [17,18,19,20,21] with a common fine-tuned architecture for TLD detection and conducts ablative research to determine which DL model performs the best.

A brief summary of earlier work addressing the issue of disease detection in tomatoes is given in Section 2. The EfficientNetB5 architecture suggested for the TLD diagnosis, classification, and the XAI approach is described in depth in Section 3. Section 4 provides evaluation metrics, and Section 5 provides a description of the experimental findings. Real-time sensation to enhance the model confidence is covered in Section 6, and Section 7 concludes by offering recommendations for further investigation.

## 2. Related Works

DL models have made significant advances in a variety of fields including, but not limited to, deep fakes [22,23], satellite image analysis [24], image classification [25,26], the optimization of artificial neural networks [27,28], the processing of natural language [29,30], fin-tech [31], intrusion detection [32], steganography [33], and biomedical image analysis [14,34]. CNNs have recently surfaced as one of the most commonly used techniques for plant disease identification [35,36].

By removing the constraints brought on by poor illumination and homogeneity in complicated environment scenarios, several works have concentrated their efforts on recognizing characteristics, while some authors have introduced real-time prediction [37,38]. For instance, research has been performed using DL models with the advancement of XAI techniques to develop a disease detection system with the major objective of pinpointing the disease and identifying the major areas of the plant and their parts that contribute to the classification [16,39].

The PlantVillage (PV) [40] dataset is a publicly available resource that contains images of various plant leaves with a range of disorders, including a tomato leaf disorder (TLD) [38]. This dataset has been used in multiple research works, including the following: Zhao et al. [41] achieved classification of TLDs using a multi-class feature-extraction approach. The residual block and the attention strategies were both integrated into the model, which was built on a deep CNN model. The model outperformed various deep-learning models with an overall accuracy of 99.24%.

Using the same image set, Bhujel et al. [42] examined the effectiveness of identifying various tomato diseases using a lightweight DL model. To enhance the performance of the model, a lightweight CNN method was combined with a number of attention strategies. The study explored the network architecture, performance, and computational complexity for the TLD dataset. The results showed improving classification accuracy upon building the compact and computationally efficient model with an accuracy of 99.69%.

TLD categorization was suggested by Ozbılge et al. [43] as an alternative to the well-known pre-trained knowledge-transferred ImageNet deep-network model and the compact deep-neural-network design with only six layers. The model’s performance on the PVdataset was tested using a number of statistical methods, and an accuracy of 99.70% was achieved. Antonio et. al. [44] suggested the use of a custom CNN-based architecture, which achieved a training accuracy of 99.99%, validation accuracy of 99.64%, precision of 99.00%, and a F1-score of 99.00% with the PV tomato dataset. With regard to the classification of the nine tomato illnesses, the recall metric had a value of 0.99.

The PlantVillage dataset was also utilized by Suryawati et al. [45] to train a model using Alexnet, GoogleNet, and VGGNet, which achieved test accuracies of 91.52%, 89.68%, and 95.25%, respectively. Transfer learning was used by Hong et al. [46] to reduce the quantity of training data needed, the amount of time typically required, and the cost of computation. Five deep-network topologies—Xception, Resnet50, MobileNet, ShuffleNet, and Densenet121—were employed to glean features from the 10 various tomato leaf disorders form the PV dataset. During the experiment, network architectures with various learning rates were contrasted. ShuffleNet had a recognition accuracy of 83.68%, whereas DenseNet and Xception had accuracies of 97.10% when the parameters were at their highest.

Vijay et al. [47] used CNN and K-nearest neighbor (KNN) models in their classification of tomato leaf disorders using the PlantVillage dataset, while LIME was used to provide explainability for the predictions made by each model. The CNN model performed better than the KNN model when used to detect leaf disease. The accuracy, precision, recall, and F1-score of the CNN model were 98.5%, 93%, 93% and 93%—all greater than those of the KNN model, which only managed to reach values of 83.6%, 90%, 84%, and 86%, respectively. Noyan et al. [48] claimed that the PlantVillage dataset is biased through the association of the background color to specific TLDs with an accuracy of around 40% for classification based on the use of background pixels only. However, Mzoughi et al. [49] demonstrated bias in the PV dataset, with image background colour associated with disease class. Additionally, they also showed improved identification outcomes, particularly in the setting of pictures with complicated backgrounds.

Kaur et al. [50] used an EfficientNetB7 model to examine leaf diseases of grape plants from the PlantVillage dataset. For the purpose of extracting the most important characteristics, the fully connected layer was created. The variance approach was then used to exclude extraneous features from the feature extractor vector. The logistic regression approach was then used to minimize the characteristics that had achieved a classification precision of 98.7%.

## 3. Materials and Methods

### 3.1. Dataset and Pre-Processing

TLD images of 10 distinct categories (healthy, bacterial spot, early blight, Septoria leaf spot, late blight, leaf mold, two-spotted spider mite, mosaic virus, target spot, and yellow leaf curl virus) were collected from the publicly available Kaggle dataset [51]—a total of 11,000 images distributed evenly among 10 classes in groups of 1100 each. The dataset was split into the ratio of 90:10 for the training and test sets. Furthermore, 10% of the training set was used as a validation set.

For better computational efficiency, each image was resized to 200 × 250. For faster convergence and to prevent the model from over and under fitting, images were shuffled relative to the position. The training samples were horizontally flipped, rotated (by a range of 20 degrees), zoomed (by a 0.2 range), and shifted (the width and height by 0.2). Figure 1 displays sample TLD images from the dataset for individual categories.

### 3.2. Proposed Method

The two main components of the proposed DL-XAI framework for TLD identification are TLD classification and explanation based on XAI algorithms. The conceptual structure of the proposed model is shown in Figure 2.

#### 3.2.1. EfficientNetB5

In order to increase the efficiency and accuracy of the model, the authors in [52] introduced EfficientNet with a novel scaling strategy that evenly increases the network’s depth, breadth, and resolution. The basic efficientNet consists of three types of blocks: stem, body, and final blocks. Based on the composition of the body (keeping the stem and final blocks the same in each variant), EfficientNet has different variants, such as EffiecientNetB0 and EffiecientNetB1. The body of each version of EffiecentNet consists of five modules, where each modules has depth-wise convolution, batch normalization, and activation layers [53].

Considering the comparative analysis from Keras [54] on the basis of size, top-one and top-five accuracy, and parameters and depth, the EfficientNetB5 model was selected as the base model in this study. To augment the base model [52], batch normalization, dense and dropout layers were added on top. Finally, a softmax layer was added with 10 dense units for TLD classification. Regularization penalties, such as a kernel regularizer (L2 regularization = 0.016), activity regularizer (L1 regularization = 0.06), and bias regularizer (L1 regularization = 0.06), were applied on a per-layer basis. A dropout rate of 0.4 and softmax activation was used on the output layer. A summary of the proposed model is shown in Table 1.

#### 3.2.2. Explainable AI

Traditional evaluation metrics do not adequately reflect the steps taken by the AI system to arrive at an output and do not allow for interpretation of the outcome. Therefore, an AI algorithm must properly describe how it arrived at its result. Particularly in domains involving high-stake decision-making, there has been an upsurge in demand for explainability of deep-learning-based systems [55]. Professionals may more easily understand the data from the DL model and use this understanding to make a quick and accurate diagnosis of a specific type of TLD. For this, LIME, and GradCAM, two commonly used XAI algorithms, are used in this study.

A binary vector ($x^{\prime }$∈ {0, 1}^*d*${}^{\prime }$^) denoting the “presence” or “absence” of a continuous patch of super-pixel was used for the explainable details of denoting an instance (*x ∈ R^d^*) using LIME. For our model, *g ∈ G* with domain {0, 1}*^d${}^{\prime }$^* was used to present the information visually. *g* represents the absence/presence of the interpretable components. Every instance of *g ∈ G* was not enough to encapsulate the explanation; as a result, Ω(*g*) was used to measure the complexity of the explanation.

To belong in any of the 10 classes for the model *f: R^d^→R*, *f(x)* was the probability of *x*, and π_*x*_(*z*) was used as a proximity measure between an instance *z* to *x* to define locality around *x*. The fidelity function, *ℑ(f,g,π_x_)*, was used as a measure of how unfaithful *g* was in approximating *f* in the locality defined by *π_x_*. When Ω(*g*) was as low as possible to increase the interpretations, the fidelity function was also minimized. The explanation generated by LIME can be summarized as Equation (Equation 1).
(1)$$\xi \left(x\right)=\underset{g\in G}{\arg\;\max}ℑ\left(f,g,\pi_{x}\right)+\Omega \left(g\right)$$

GradCAM is a technique that calculates the gradient of a differentiable output, such as a class score, with respect to the convolutional features of a chosen layer. It is mostly used for image classification tasks but can also be applied to semantic segmentation. In the proposed model, the softmax layer generates a score for each class and each pixel to assist in the semantic segmentation task. Equation (Equation 2) explains GradCAM mapping for a specific class *C* with *N* pixels and *A^K^* as a feature map.
(2)$$M^{c}=ReLU\left(\sum_{K}\alpha_{c}^{K}A^{K}\right)$$
where,
(3)$$\alpha_{c}^{K}=\frac{1}{N}\sum_{i,j}(\frac{dy^{c}}{dA_{i,j}^{K}})$$

### 3.3. Implementation Details

Using Keras [54] in Python [56], the suggested DL model and XAI algorithms were implemented in Google Colab [57] with an NVIDIA K80 graphics processing unit and 12 GB of RAM. Python (version 3.7) and Keras (version 2.5.0), which collaborate with the TensorFlow (version 2.5.0) framework, was provided by Google Colab as a runtime platform. During the training and validation of the proposed model, two callbacks were implemented. The first callback was used to monitor the validation loss and learning rate reduction with a factor of 0.5. The second callback was used for early stopping by recovering the best points throughout the course of four epochs. Furthermore, to prevent the model from over-fitting, both callbacks were imposed within 50 epochs.

## 4. Evaluation Metrics

To evaluate the model’s ability in recognizing TLD images, standard performance metrics, such as the precision (Equation 4), recall (Equation 5), F-score (Equation 6), and accuracy (Equation 7), were applied [58].
(4)$$Precision\left(P\right)=\frac{T_{P}}{T_{P}+F_{P}}$$
(5)$$Recall\left(R\right)=\frac{T_{P}}{T_{P}+F_{N}}$$
(6)$$F-score\left(F\right)=2\times \frac{P\times R}{P+R}$$
(7)$$Accuracy\left(A\right)=\frac{T_{P}+T_{N}}{T_{P}+T_{N}+F_{P}+F_{N}}$$
where, respectively, “$T_{P}$”, “$T_{N}$”, “$F_{P}$”, and “$F_{N}$” stand for “true positive”, “true negative”, “false positive”, and “false negative”. Along with measuring the model’s performance, assessments for individual classes were enacted using the confusion matrix.

## 5. Results and Discussion

### 5.1. Comparison with Existing Pre-Trained DL Models

In order to compare the performance of the fine-tuned EfficientNetB5 model with other existing transfer-learning approaches using pre-trained Dl models, we chose MobileNet [17], Xception [19], VGG16 [18], ResNet50 [20], and DenseNet121 [21] and implemented transfer learning on the TLD dataset under the same implementation details as described in Section 3.3. Since the existing DL models range from heavy-weight (high numbers of trainable parameters) to light-weight (lower number of trainable parameters), we opted to cover both kind of models where VGG-16, ResNet50, etc. represent the heavy-weight models, while MobileNet represents the lightweight model). The proposed DL model with EfficientNetB5 outperformed the other models in terms of accuracy and loss (Table 2).

More specifically, the proposed DL model with EfficientNetB5 achieved the highest test accuracy of 99.07% in comparison to the test accuracy of the other models, such as MobileNet (94.00%), Xception (95.32%), VGG16 (93.35%), ResNet50 (96.03%), and DenseNet121 (96.30%). In comparing the performance of the proposed DL model with existing pre-trained DL models, EfficientNetB5 had better performance (by 2.77% accuracy) compared to the second-best performing model (DenseNet121). Similarly, among the comparison cohort, VGG16 was the least performing model (with test accuracy of 93.35%), which is significantly lower (by 5.72%) than the performance of the proposed DL model with EfficientNetB5.

### 5.2. Model Explanation with EfficientNetB5

Conventional statistical validation protocols were considered, encompassing the metrics of model loss and accuracy across the training, validation, and test sets as well as the precision, F1-score, and recall. In terms of model training, a predetermined termination criterion of 50 epochs was imposed. Figure 3 shows the 10th-fold results, and the model achieved an average training accuracy of 99.84% ± 0.10%and validation accuracy of 98.28% ± 0.20% (refer to Table 3).

On a test set that was not shown to the model during training, the model was further assessed to achieve a test accuracy of 99.07% ± 0.38% and a test loss of 0.20 ± 0.03. The results from the confusion matrix (Figure 4) show that bacterial spot, late blight, leaf mold, spider mite, and yellow curl had a single instance predicted wrong, whereas the early bright, Septoria leaf spot, target spot, mosaic virus, and healthy categories were all predicted correctly. The precision, F1-scores and recall of the individual categories are shown in Table 4.

We evaluated the AUC ROC scores for each class to assess the efficiency of the proposed model (Figure 5). The average AUC ROC score across all classes was 1.0—demonstrating the model’s accuracy in correctly classifying instances. The CNN performed well for all courses as per the AUC ROC values for individual classes. These outcomes show how successfully the model handled the multi-class categorization issue.

### 5.3. Model Explanation with XAI

#### 5.3.1. GradCAM

The GradCAM technique was employed to discern the salient regions of TLD images that were instrumental in the classification process by leveraging the spatial information retained by convolutional layers. To evaluate the efficacy of the proposed visual explanation techniques, a comprehensive examination of individual TLD samples from every category was conducted, which involved visual inspection of the heatmaps generated by the methodologies. The heatmap is shown in Table 5 as a result.

Regarding the results of GradCAM for bacterial spot (Table 5, first row), the middle section of the leaf holds the greatest impact on the classification and heatmap positions on the center section. On the other hand, for early blight, the right portion in the image shows as more infectious, and the heatmap is plotted accordingly. For leaf mold, GradCAM is concentrated on the yellow section of the leaf. However, in the GradCAM heatmap for the class target spot (Table 5, fifth row), some background portion of the leaf images is also highlighted with the majority of the gradient concentrated on the center of the images. This shows that the model is also considering the background weight while making decisions; however, the impact of the background is negligible as seen while testing the model on the independent test images (refer to Section 6).

#### 5.3.2. LIME

A matrix was created with 150 rows of randomly generated ones and zeros, where the columns were made up of superpixels. The matrix was split with a ratio of 0.2, and it was perturbed using a kernel size of 3 by 3 and a maximum distance of 100 units. The perturbations were applied to the top 20 numerical features, which were adjusted to align with the averages and standard deviation of the training data using a normal (0, 1) sampling method and by undoing the mean-centering and scaling operations. A robust binary feature was crafted using training-distribution-based sampling to generate categorical features. This method was executed to develop a feature that is unequivocally assigned the value of 1 when it corresponds to the instance being described. The individual segmentation of TLD is shown in Table 5.

Considering bacterial spot from Table 5, first row, the leaf holds the bacterial spot. The center portion of the leaf has the greatest impact on the classification, and LIME segments the portions accordingly. For early blight, in examining the LIME output, we can see that EfficientNetB5 activates toward the right section of the leaf. Accordingly, for leaf mold, the yellow section of the leaf is indicated as an important characteristic used by the network to classify the leaf as leaf mold.

### 5.4. Comparison with the State-of-the-Art Methods

Table 6 compares the classification performance of the proposed model with existing state-of-the-art approaches. To increase the performance’s coherence and relevance, we chose the most current disease detection models using deep-learning approaches based on TLD categorization. To serve as a comparison population, we choose a total of seven contemporary DL methods. Additionally, this comparative cohort employed both transfer learning-based DL models and custom-CNN models that were created from scratch. Only two of the four publications employed customizable CNNs with LIME as XAI. The TLD dataset was classified using the transfer learning technique in the other three studies; however, none of them employed XAI. The proposed method outperformed all other state-of-the-art methods.

## 6. Independent Validation

The use of test data independent of the training set is recommended in any model building. In this case, Noyan et al. [48] noted bias in the PlantVillage (PV) dataset associated with background color. It was, therefore, essential to impose our model on tomato leaf images that were drawn from other sources. Five random images for respective classes were collected from [59], which holds 32,535 images for 11 different tomato leaf diseases. Resized to 250 × 200, the collected images (not used to train, test, and validate the proposed model) were fed into the model, and real-time sensational validation for individual categories was calculated. The predicted probability for the sample images and GradCAM results are tabulated in Table 7. It is seen that the GradCAM results identify the infected section, and no spots are present inside the healthy portion of the leaf. Table 8 shows the class-wise accuracy. Both leaf mold and spider mite achieved 80% accuracy, whereas the rest of the categories achieved 100% with an average accuracy of 96%.

## 7. Conclusions and Future Work

In this study, we suggested an explanation-generation (XAI) framework together with a CNN model for categorizing TLD diseases into nine separate classes. Transfer learning with EfficientNetB5 was used to create the BotanicX-AI to deliver explanation-driven findings, and GradCAM and LIME produced a detailed explanation of the results. The proposed work was compared to existing pre-trained DL models for TLD detection with a common fine-tuned architecture, and ablative research to find the optimal DL model was conducted.

The test and training accuracies obtained with an XAI-based CNN model to predict TLD were 99.07% ± 0.38% and 99.84% ± 0.10%, respectively. The explanations generated with both GradCAM and LIME were able to identify the specific region that contributed to the categorization of TLD. The suggested model shows that XAI and EfficientNetB5 generated admissible explanations of the outcomes with high classification accuracy.

In contrast, we observed that GradCAM failed to identify regions of an image used by model prediction due to the gradient-averaging step. HiResCAM is a generative adversarial network (GAN), and global model-specific explanation AI, such as kernel SHapley Additive exPlanations (SHAP), are recommended as potential alternatives. Trials of these procedures are recommended. The further development of the PlantVillage dataset for tomato leaf disorders is also recommended to address the issue of bias associated with the background as raised by [48]. Furthermore, as the proposed model is based on the EfficientNetB5 architecture, it may be further evaluated with datasets that have more information and images.

## Figures and Tables

**Figure 1 jimaging-09-00053-f001:**
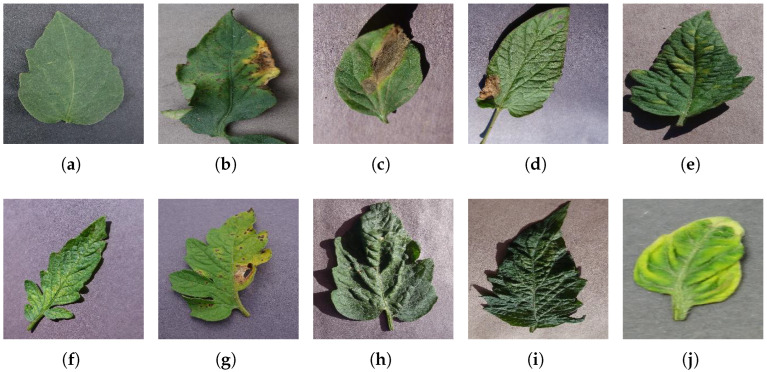
Distinct sample images from the dataset for individual diseases. (**a**–**j**) denote “healthy”, “bacterial spot”, “early blight”, “late blight”, “leaf mold”, “mosaic virus”, “Septoria leaf spot”, “target spot”, “two spotted spider mite”, and ”yellow leaf curl virus” classes, respectively.

**Figure 2 jimaging-09-00053-f002:**
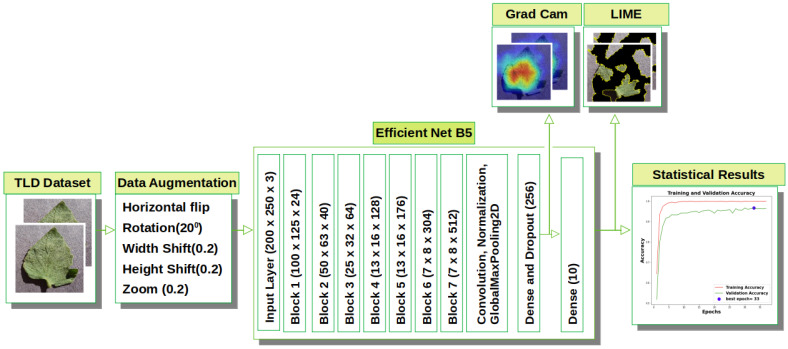
Proposed high level conceptual architecture of explanation—driven DL model (BotanicX-AI) for TLD detection.

**Figure 3 jimaging-09-00053-f003:**
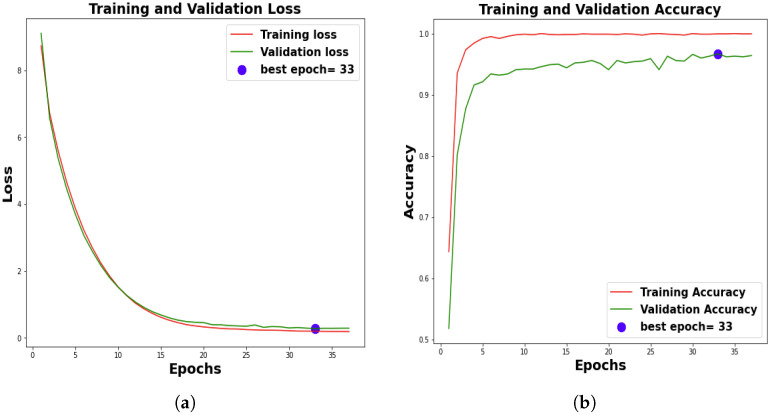
Training and validation results. (**a**) 99.84% ± 0.10% average training accuracy and 99.07% ± 0.38% average validation accuracy over 10 folds. (**b**) 0.18 ± 0.01 training loss and 0.24 ± 0.02 validation loss.

**Figure 4 jimaging-09-00053-f004:**
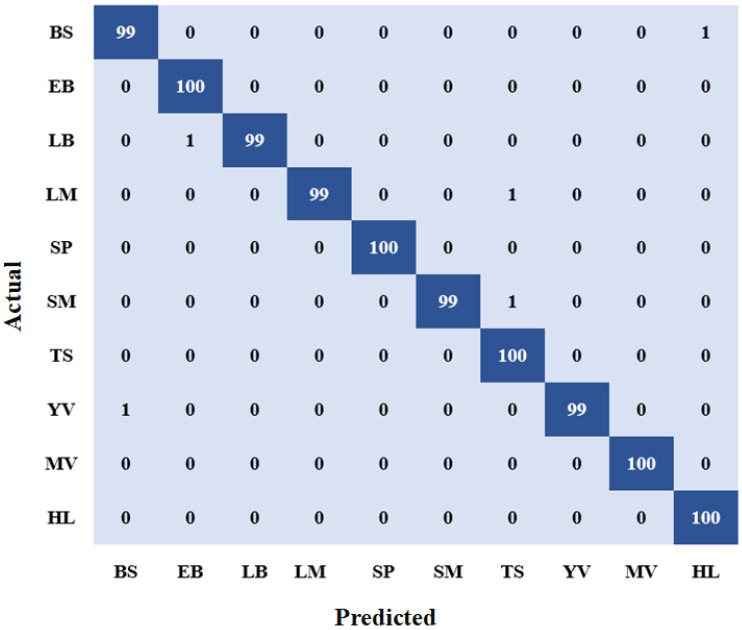
Confusion Matrix. ‘BS’, ‘EB’, ‘LB’, ‘LM’, ‘SP’, ‘SM’, ‘TS’, ‘YV’, ‘MV’, and ‘HL’ stand for bacterial spot, early blight, late blight, leaf mold, Septoria leaf spot, spider mite, target spot, yellow curl virus, mosaic virus, and healthy leaves, respectively.

**Figure 5 jimaging-09-00053-f005:**
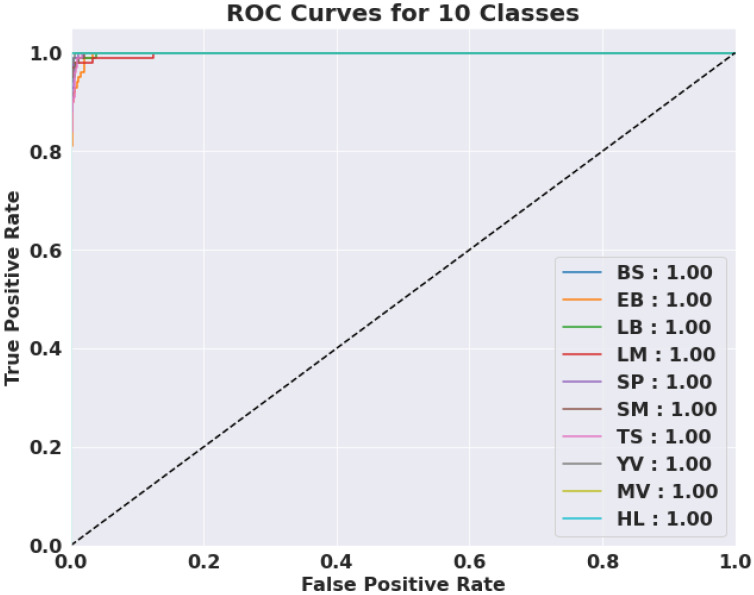
The AUC-ROC results of the proposed model with an AUC score of 1.0.

**Table 1 jimaging-09-00053-t001:** Summary for the proposed model. Note that “Param. #” denotes the number of parameters in the respective layer.

Layer (Type)	Output Shape	Param. #
efficientnetb5 (Functional)	(None, 2048)	28,513,527
batch_normalization (BatchNormalization)	(None, 2048)	8192
dense (Dense)	(None, 256)	524,544
dropout (Dropout)	(None, 256)	0
dense (Dense)	(None, 10)	2570
Total parameters: 29,048,833		
Trainable parameters: 28,871,994		
Non-trainable parameters: 176,839		

**Table 2 jimaging-09-00053-t002:** Analysis of the suggested model in comparison to the MobileNet, Xception, VGG16, ResNet50, and DenseNet121 models. Training accuracy, training loss, validation accuracy, validation loss, test accuracy, and test loss are abbreviated as ‘TA’, ‘TL’, ‘VA’, ‘VL’, ‘TsA’, and ‘TsL’, respectively.

DL Model	TA	TL	VA	VL	TsA	TsL
MobileNet	99.90	0.1957	96.10	0.2871	94.00	0.9012
Xception	99.90	0.1836	97.70	0.3494	95.32	0.3921
VGG16	83.20	1.0951	79.10	1.1589	93.35	2.39
ResNet50	99.95	0.2204	97.80	0.2702	96.03	0.3569
DenseNet121	99.90	0.2235	96.80	0.2972	96.30	0.3038
This work (EfficientNetB5)	99.84	0.18	99.07	0.24	99.07	0.20

**Table 3 jimaging-09-00053-t003:** Ten-fold training, testing, and validation performance in percentages; training accuracy (TA), validation accuracy (VA), training loss (TL), validation loss (VL), test accuracy (TsA), and test loss (TsL).

	TA	TL	VA	VL	TsA	TsL
K1	99.85	0.1842	98.00	0.2263	99.10	0.1980
K2	99.95	0.1851	97.90	0.2392	99.20	0.1874
K3	99.82	0.1738	98.34	0.2195	99.21	0.1890
K4	99.90	0.1900	98.50	0.2830	98.30	0.2630
K5	99.85	0.1865	98.60	0.2515	99.70	0.2497
K6	99.85	0.1981	98.40	0.2593	99.10	0.2180
K7	99.90	0.1331	98.20	0.2480	98.90	0.1914
K8	99.70	0.1567	98.30	0.2131	98.65	0.1856
K9	99.60	0.2011	98.37	0.2211	99.10	0.1760
K10	99.95	0.1883	98.20	0.2187	99.50	0.1754
μ±σ	99.84 ± 0.10	0.18 ± 0.01	98.28 ± 0.20	0.24 ± 0.02	99.07 ± 0.38	0.20 ± 0.03

**Table 4 jimaging-09-00053-t004:** Class-wise prevision, recall, F1-score, and support (number of samples) for each TLD class on the test dataset.

	Precision	Recall	F1-Score
Bacterial spot	0.9900	0.9900	0.9900
Early blight	0.9901	1.0000	0.9950
Late blight	1.0000	0.9900	0.9950
Leaf mold	1.0000	0.9900	0.9950
Septoria leaf spot	1.0000	1.0000	1.0000
Spider mite	1.0000	0.9900	0.9901
Target spot	0.9804	1.0000	0.9901
Yellow curl virus	1.0000	0.9900	0.9950
Mosaic virus	1.0000	1.0000	1.0000
Healthy	0.9901	1.00	0.9955
Accuracy			0.9950
Macro Avg	0.9951	0.9950	0.9950
Weighted Avg	0.9951	0.9950	0.9950

**Table 5 jimaging-09-00053-t005:** Explainable AI result interpretations for TLD.

Category	Leaf	GradCAM	LIME
Bacterial spot	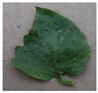	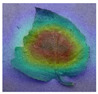	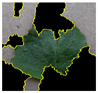
Early blight	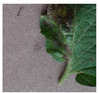	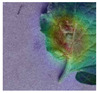	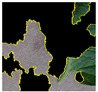
Late blight	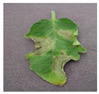	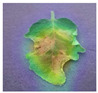	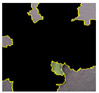
Leaf mold	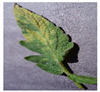	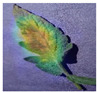	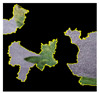
Septoria spot	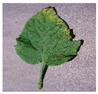	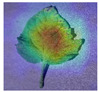	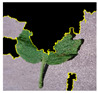
Spider mite	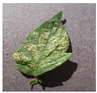	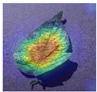	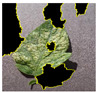
Target spot	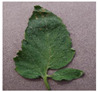	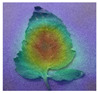	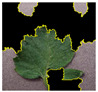
Yellow leaf	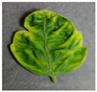	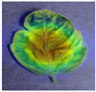	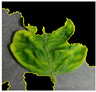
Mosaic virus	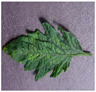	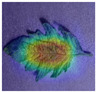	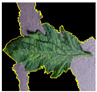

**Table 6 jimaging-09-00053-t006:** Comparison of our model with the state-of-the-art methods on the TLD dataset.

Ref.	Method	Accuracy (%)	XAI
[41]	CNN with attention module	99.24	No
[50]	EfficientNet B7	98.7%	No
[45]	Alexnet, GoogleNet, and VGGNet	91.52%, 89.68%, and 95.25%	No
[43]	Compact-CNN	99.70%	GradCAM
[12]	Deep-CNN	98.49%	No
[46]	Densenet Xception	97.10%	No
[47]	XAI-CNN	98.5%	LIME
This work	EfficientNetB5	99.84% ± 0.10%	GradCAM, LIME

**Table 7 jimaging-09-00053-t007:** Real-time results.

Image	Predicted Probability	GradCAM
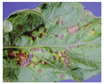	Bacterial spot (80.56%)	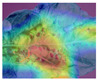
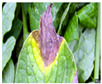	Early blight (90.26%)	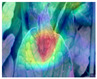
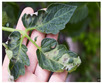	Late blight (91.60%)	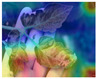
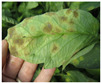	Leaf mold (93.42%)	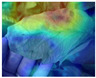
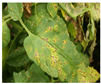	Septoria leaf spot (90.35%)	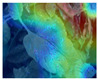
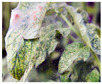	Spider mite (88.41%)	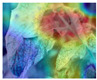
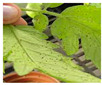	Target spot (96.87%)	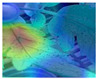
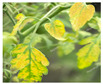	Yellow leaf (87.58%)	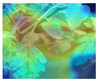
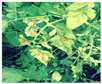	Mosaic virus (85.33%)	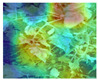
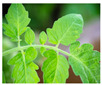	Healthy leaf (95.19%)	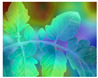

**Table 8 jimaging-09-00053-t008:** Class-wise accuracy for the independent dataset. ‘BS’, ‘EB’, ‘LB’, ‘LM’, ‘SP’, ‘SM’, ‘TS’, ‘YV’, ‘MV’, and ‘HL’ stand for bacterial spot, early blight, late blight, leaf mold, Septoria leaf spot, spider mite, target spot, yellow curl virus, mosaic virus, and healthy leaves, respectively.

Class	Accuracy
BS	100%
EB	100%
LB	100%
LM	80%
SP	100%
SM	80%
TS	100%
YV	100%
MV	100%
HL	100%
Average	96%

## Data Availability

Publicly available tomato leaf diseases dataset, https://www.kaggle.com/datasets/kaustubhb999/tomatoleaf (Accessed on 3 July 2022).

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
