# Peer review of "BotanicX-AI: Identification of Tomato Leaf Diseases Using an Explanation-Driven Deep-Learning Model"

_2313-433X, 2023, doi:10.3390/jimaging9020053_

Round 1

Reviewer 1 Report

Comment 1: The PlantVillage dataset has serious bias issues. In the paper "Uncovering bias in the PlantVillage dataset" (https://arxiv.org/abs/2206.04374), they trained a disease classification model for all 38 classes on eight background pixels. For an unbiased dataset, this model should have performed around random guess accuracy (2-3%). Instead, it achieved 49% classification accuracy, which indicates significant bias.

In light of this, ~99% of accuracies reported in the BotanicX-AI paper should be justified.

Comment 2: In section 6. Real Time Sensation, the authors tested their model on other datasets, which is an important step towards the justification requested in comment 1, however, they do not report any discrimination metric. Table 8 does not prove that the model will work reliably on other datasets; it merely says the model is looking at the leaves. But what is accuracy? What is the reason for omitting these metrics?

Comment 3: Authors mention they have 1100 images per class. But the PlantVillage has less than 1100 images for 3 out of 10 classes (early blight, leaf mold, and mosaic virus). Can you please explain this? Where did the additional data come from?

Comment 4: The model development part of the paper looks fine. In section 3.3, it seems to me a single callback is used (Early stopping), but the authors mention "both callbacks" (page 5, line 163). Can you please clearly define the two? Also, the correct terminology is early stopping, not early callback termination.

Author Response

Please find the attached response to reviewer

Reviewer 2 Report

Dear Authors,

In this manuscript a pre-trained deep learning model is used for tomato leaf disease detection which is useful in agricultural sector. The proposed method is used to identify healthy with 9 tomato diseases. All the parts of the manuscript are well-written with a proper methodological approach and justification.

There are some minor revisions proposed and one question for the authors: 

1) The authors are advised to add more relevant references in introduction and related works sections.

2)  page 2, line 44 "computer visions" -> "computer vision"

3) In "Related works" -> "DL models have made significant .. image analysis [19]" add more references in each field.

3) Replace all "authors" in "Related Works" section with authors' names. e.g. Page 2 Line 81 "authors in [32]" or "Authors in [33]" as it is easier for the reader.

4) In Page 3 change the "Table 1" to "Figure 1.a..k"

5) In Figure 3 add "healthy" before the word "respectively"

6) Please emphasize and enrich the contributions of the paper in conclusion section.

Question about the authors

Did you check a different ratio between training:testing:validation images?

Author Response

Please find the attached response to the reviewer.

Round 2

Reviewer 1 Report

Thank you for your efforts in addressing my comments. I believe the manuscript improved significantly. However, there is one remaining issue. In my first review, I suggested including a discrimination metric such as accuracy on the independent dataset. Instead, you reported "confidence levels".  This is not a discrimination metric. It does not tell us how good/bad the model is doing. Please report the accuracy of the model on the whole independent dataset.

Author Response

Please find the attached Reviewer report. 
